# Diagnostic accuracy of point-of-care ultrasound for pulmonary tuberculosis: A systematic review

Jacob Bigio[1,2]*, Mikashmi Kohli[2,3], Joel Shyam Klinton[2,3], Emily MacLean[2,3], Genevieve Gore[4], Peter M. Small[5], Morten Ruhwald[6], Stefan Fabian Weber[7], Saurabh Jha[8], Madhukar Pai[2,3]

1 Research Institute of the McGill University Health Centre, Montreal, Canada, 2 McGill International TB Centre, Montreal, Canada, 3 Department of Epidemiology, Biostatistics and Occupational Health, McGill University, Montreal, Canada, 4 Schulich Library of Physical Sciences, Life Sciences, and Engineering, McGill University, Montreal, Canada, 5 Renaissance School of Medicine at SUNY, Stony Brook, NY, United States of America, 6 Foundation for Innovative New Diagnostics, Geneva, Switzerland, 7 Department of Infectious Diseases, University of Heidelberg, Heidelberg, Germany, 8 Department of Radiology, Hospital of the University of Pennsylvania, Philadelphia, PA, United States of America

* jacob.bigio@affiliate.mcgill.ca

**Data Availability Statement:** All relevant data are within the paper and its Supporting Information files.

## Abstract

The advent of affordable, portable ultrasound devices has led to increasing interest in the use of point-of-care ultrasound (POCUS) for the detection of pulmonary TB (PTB). We undertook a systematic review of the diagnostic accuracy of POCUS for PTB. Five databases were searched for articles published between January 2010 and June 2020. Risk of bias was assessed using QUADAS-2. Data on sensitivity and specificity of individual lung ultrasound findings were collected, with variable reference standards including PCR and sputum smear microscopy. Six of 3,919 reviewed articles were included: five in adults and one in children, with a total sample size of 564. Studies had high risk of bias in many domains. In adults, subpleural nodule and lung consolidation were the lung ultrasound findings with the highest sensitivities, ranging from 72.5% to 100.0% and 46.7% to 80.4%, respectively. Only one study reported specificity data. Variability in sensitivity may be due to variable reference standards or may imply operator dependence. There is insufficient evidence to judge the diagnostic accuracy of POCUS for PTB. There is also no consensus on the optimal protocols for acquiring and analysing POCUS images for PTB. New studies which minimise potential sources of bias are required to further assess the diagnostic accuracy of POCUS for PTB.

## Introduction

Tuberculosis (TB) is one of the top ten causes of death worldwide, with an estimated 10 million new cases leading to 1.4 million deaths in 2019 [1]. The End TB Strategy targets a 90% reduction in TB incidence rate and a 95% reduction in TB deaths by 2035 and the early diagnosis of

**Funding:** The authors received no specific funding for this work.

**Competing interests:** The authors have declared that no competing interests exist.

TB is a key component of the strategy [2]. However, analyses of cascades of care from high burden countries show substantial patient attrition at the diagnosis stage [3,4].

Chest x-ray (CXR) is an established systematic screening tool for active pulmonary TB (PTB) and an established triage tool to identify adults with presumptive active PTB to refer for confirmatory testing with culture or a molecular test [5,6]. CXR has a sensitivity of 87% and a specificity of 89% for adult PTB [7]. Diagnosis of PTB in children is more challenging, given the paucibacillary nature of the disease and difficulty in obtaining clinical samples [8]. As such, CXR is frequently used alongside clinical symptoms to make a presumptive diagnosis of PTB in the absence of bacteriological confirmation [5] or to classify cases as unconfirmed TB in conjunction with a number of criteria including immunologic evidence of *Mycobacterium tuberculosis* infection and a positive response to anti-TB therapy [8].

Despite its proven utility for PTB, however, CXR hardware is expensive and the availability of CXR is limited in many high TB-burden, low-resource settings due to scarcity of both equipment and skilled radiological staff to operate and interpret the images. [9,10]. Artificial intelligence (AI)-assisted interpretation of CXRs recently obtained WHO policy recommendation [11] and may become widespread in the coming years, reducing the requirement for skilled staff, but CXR hardware cost remains a huge barrier for access [12–14].

By contrast, point-of-care ultrasound (POCUS) devices are inexpensive, easily portable to rural centres, do not result in exposure to ionising radiation and do not require radiological staff [9,10], making their use attractive for practical reasons in low-resource settings. The term POCUS has been used in different ways in the literature but is generally defined as an ultrasound exam performed and interpreted in real-time by a single non-radiologist operator [15–17].

POCUS devices have been used for the diagnosis of several infectious diseases in low- and middle-income settings [17], notably with the focused assessment with sonography for human immunodeficiency virus (HIV)-associated extrapulmonary TB (FASH) protocol [18,19]. Lung ultrasound (LUS) has also been used successfully in the diagnosis of adult pneumonia, with meta-analyses suggesting it has similar or higher sensitivity and specificity to CXR [20–22]. There is therefore reason to suspect that POCUS may be a suitable imaging modality for PTB and there has been increasing interest in its use in both adults and children, especially as POCUS becomes cheaper and more widely used in clinical medicine [23,24].

Unlike CXR, it is unclear whether POCUS images are suitable for AI-assisted interpretation. However, early work has been undertaken on the AI-assisted interpretation of LUS findings for paediatric pneumonia and COVID-19 [25–27], suggesting it may one day become a possibility for POCUS for PTB. Either way, if POCUS could be shown to have similar diagnostic accuracy to CXR for PTB, it could be a valuable diagnostic tool given its low hardware costs, ability to reach primary care, and near-immediate provision of results.

The WHO's target product profile (TPP) for a triage tool for PTB suggests a minimum requirement of 90% sensitivity and 70% specificity with a price per test of less than $2 and a time to result of less than 30 minutes [28]. We aimed to evaluate whether POCUS meets these diagnostic accuracy requirements, or whether it would make a suitable systematic screening tool, for which the WHO has not released a TPP but the diagnostic accuracy of CXR may provide a benchmark.

A 2018 systematic review, which included studies published up to 2016, found no studies with data on the diagnostic accuracy of LUS for PTB [29]. This systematic review provides an updated picture, and aims to evaluate the diagnostic accuracy and reproducibility of POCUS for PTB in both adults and children.

## Methods

### Search strategy

In consultation with a librarian (GG), a search strategy was developed to identify relevant literature in MEDLINE (Ovid), Embase (Ovid), SCI-EXPANDED and ESCI (Web of Science), CENTRAL (Cochrane Library) and SCOPUS using terms relating to tuberculosis, ultrasound and either screening or diagnosis [S1 Appendix]. The search was limited to articles published in English or French from January 1, 2010 to June 1, 2020. No review protocol was registered.

### Study selection

Two reviewers (JB and JSK) independently conducted the title/abstract screening of all articles and two reviewers (JB and MK) independently conducted the full text screening of all included titles/abstracts. Articles were assessed using pre-defined inclusion and exclusion criteria, with conflicts resolved through discussion between the pairs of reviewers.

Studies that assessed the diagnostic accuracy or reproducibility of transthoracic ultrasound for PTB disease were included. Studies that assessed the use of endoscopic ultrasound and ultrasound-guided biopsies, as well as those evaluating people with presumed extrapulmonary TB (EPTB) or latent TB, were excluded. Quantitative observational studies, mixed methods studies with a quantitative component and intervention studies, including conference presentations and abstracts, were included. Qualitative studies, modelling studies and economic evaluations were excluded. Studies in adults (older than 15 years) were included if the ultrasound findings were assessed against a reference standard of liquid or solid culture or a molecular test (higher-quality outcome) or assessed for agreement with findings from another imaging modality (lower-quality outcome). Studies in children (younger than 15 years) were included if the ultrasound findings were assessed against any reference standard (including liquid or solid culture, molecular tests, sputum-smear microscopy, clinical reference standards, other imaging modalities or any composite reference standard). Studies not published in English or French were excluded. Grey literature was excluded, except for conference presentations and abstracts indexed in the five searched databases.

### Data extraction

Data were extracted using a standardised extraction form in Microsoft Excel [S2 Appendix]. Two reviewers (JB and MK) independently performed the extraction. Extracted data were compared and any discrepancies were resolved through consensus between the reviewers. Extracted data included: study design and patient selection methods; study location, healthcare setting and type of healthcare provider; patient demographics, including HIV status and history of TB treatment; specimen type, reference standard method; true positives, false positives, true negatives, false negatives by ultrasound finding; kappa score for reliability by ultrasound finding.

### Quality assessment

Quality assessment was conducted independently by two reviewers (JB and EM) for all included studies using the revised tool for the Quality Assessment of Diagnostic Accuracy Studies (QUADAS-2) [30]. An additional domain was added to assess the quality of reproducibility study data using the following criteria, based on the work of Mokkink et al. (2018) [31] [S3 Appendix]. Detailed guidance for the answering of the QUADAS-2 and additional reproducibility questions was pre-defined [S4 Appendix]. Disagreements were resolved through consensus between the reviewers. All assessed studies were included, regardless of the QUADAS-2 results.

## Data analysis

As the presentations of adult and childhood PTB are different [7,32], the diagnostic accuracy of ultrasound for PTB was *a priori* assumed to be different for adults and children. Data are therefore presented separately for adults (older than 15 years) and children (15 years or younger). Exact binomial 95% confidence intervals were calculated for all estimates of sensitivity and specificity. Data were not meta-analysed in adults due to heterogeneity of reference standards [Table 1]. Only one study in children was included.

# Results

## Study selection

After deduplication, 3,919 records were identified. 3,864 records were excluded after title and abstract screening. Of the remaining 55 studies, 49 were excluded after full-text review [Fig 1]. Most full-texts were excluded for not being about PTB, for not being diagnostic accuracy studies or for being reviews or editorials [S5 Appendix]. No studies were

**Table 1. Individual study characteristics for studies in adults.**

| Article | Study design | Participant Selection | Country | Setting | Specimen type for reference standard | Reference standard | Sonographer | HIV status (HIV +ve/ total) (patient subgroup) | Previous history of TB | Age range in years (patient subgroup) | Median age in years (IQR), unless stated | Gender (male/ female) (% male) |
|---|---|---|---|---|---|---|---|---|---|---|---|---|
| Agostinis 2017 [34] | Prospective cross-sectional | – | Guinea-Bissau | Regional hospital | Sputum | Clinical symptoms, AFB and CXR | – | 30/60 (50%) | – | – | 32.5 (18.1) | 27/33 (45%) |
| Babasa 2019 [35] | Prospective cross-sectional | Consecutive | Philippines | Tertiary hospital | Sputum | NAAT, AFB and CXR | Emergency physician trained in lung ultrasound | – | – | – | – | – |
| Fentress 2020 [33] | Prospective cross-sectional | Consecutive | Peru | Regional hospital | Sputum | AFB (50/ 51); or PCR/ culture (1/ 51) | General practitioners following 30 hours' training | 0/51 (0%) | – | 18–78 | Mean 33.7, SD 15.81 | 35/16 (69%) |
| Montuori 2019 [9] | Prospective cross-sectional | – | Italy | Tertiary hospital | – | AFB, PCR and solid and liquid culture (95/ 102); or clinical symptoms and CXR (7/102) | Internal medicine physician experienced in clinical ultrasonography | 11/51 (22%) (PTB) | 7/51 (14%) | 24–49 (PTB) | 34 | 37/14 (73%) |
| | | | | | | | | 17/51 (33%) (non-PTB) | 14/51 (27%) | 39–60 (non-PTB) | 49 | 30/21 (59%) |
| Wagih 2020 [36] | Prospective cross-sectional | – | Egypt | Tertiary hospital | – | AFB; or PCR (unclear proportion) | – | 25/50 (50%) | – | 21–51 (HIV +ve) | Mean 34.6, SD 8.6 | 23/2 (92%) |
| | | | | | | | | | – | 17–61 (HIV -ve) | Mean 33.9, SD 13.6 | 25/0 (100%) |

AFB = sputum smear microscopy for acid fast bacilli; CXR = chest x-ray; HIV = human immunodeficiency virus; NAAT = nucleic acid amplification test; PCR = polymerase chain reaction; PTB = diagnosed with pulmonary tuberculosis by reference standard; non PTB = not diagnosed with pulmonary tuberculosis by reference standard;– = data unavailable.

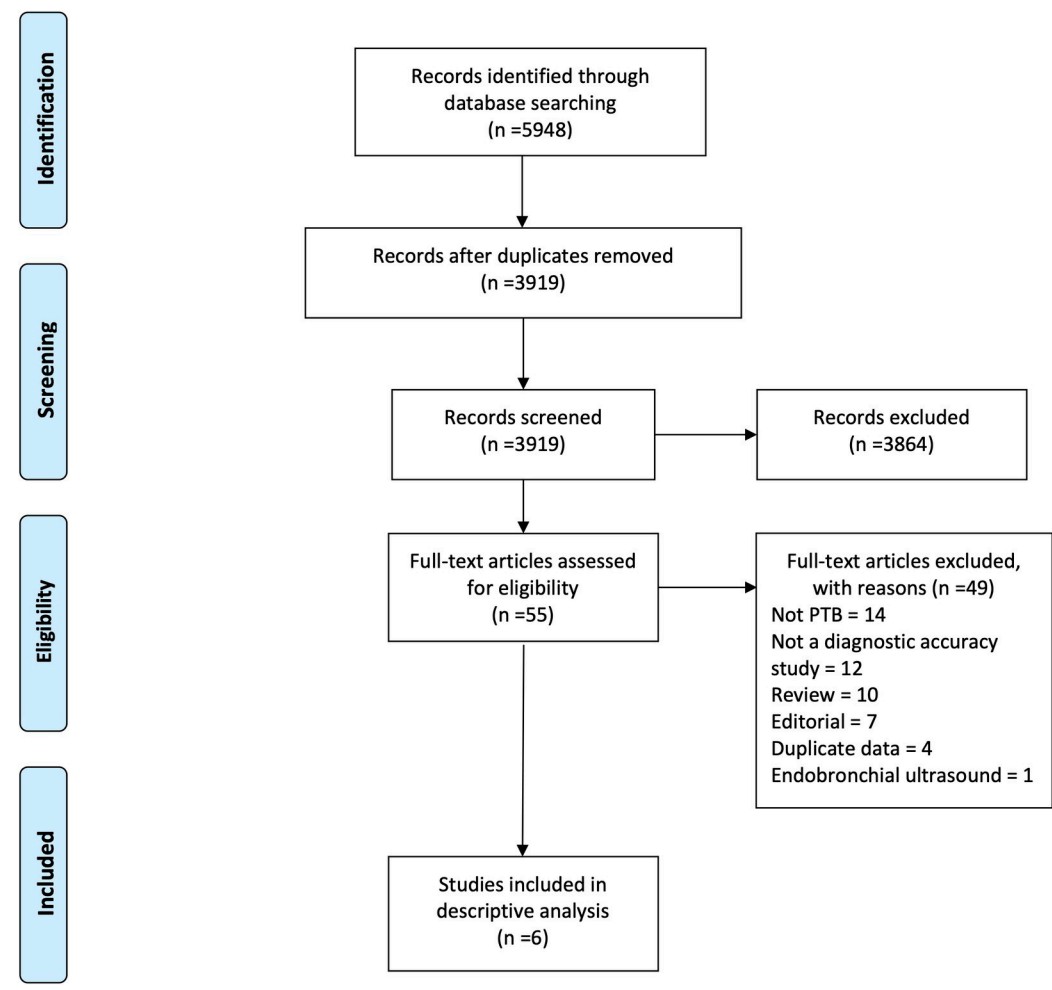

**Fig 1. PRISMA study flowchart.**

excluded for not being in English or French. Two of the six included studies were conference abstracts. Fentress and colleagues shared the data behind their abstract and preliminary versions of a manuscript which has subsequently been published as Fentress et al. (2020) [33].

Five of the six included studies were in adults [9,33–36] and one was in children [37]. Detailed study characteristics are shown in Table 1 for adults and Table 2 for children. All studies described using POCUS devices. Four (80%) of the five adult studies were in low- and middle-income countries (LMICs) and one (20%) was in a high-income country. Three (60%) were in tertiary hospitals and two (40%) were in regional hospitals. Three (60%) had sputum as the specimen type for the reference standard and two (40%) did not report these data. All studies reported different combinations of reference standards. The percentage of patients who were HIV positive ranged from 0% to 50%. Median or mean age ranged from 32.5 to 41.5 years. The percentage male ranged from 45% to 96%. The one paediatric study was in a tertiary hospital in South Africa, an LMIC. Sputum was the specimen type, with a liquid culture and PCR or clinical reference standard. 14% of patients were HIV positive. Median age was 26.6 months. The percentage male was 57%.

**Table 2. Individual study characteristics for studies in children.**

| Article | Study design | Participant selection | Country | Setting | Specimen type for reference standard | Reference standard | Sonographer | Patient category | HIV status (HIV +ve/ total) | Median age in months (IQR) | Gender (male/ female) (% male) |
|---|---|---|---|---|---|---|---|---|---|---|---|
| Heuvelings 2019 [37] | Prospective cohort study | Consecutive | South Africa | Tertiary hospital | Sputum | Liquid culture + PCR; or clinical | Clinician who attended a 4-day ultrasound training (85%); trained sonographer with 11 years of echocardiography experience (15%) | Confirmed TB | 6/40 (15%) | 48.5 (18.3–71.0) | 26/14 (65%) |
| | | | | | | | | Unconfirmed TB | 12/85 (14%) | 23.9 (13.3–43.0) | 50/35 (59%) |
| | | | | | | | | Unlikely TB | 5/45 (11%) | 23.9 (17.3–56.2) | 20/25 (44%) |

PCR = polymerase chain reaction; confirmed TB = *M. tuberculosis* detected by either culture or PCR; unconfirmed TB = clinical diagnosis for PTB but negative microbiological test result; unlikely TB = respiratory disease due to other organisms or symptoms improved without TB treatment.

## Quality assessment

Table 3 shows the quality assessments for the studies in adults. Risk of bias was high in four (80%) studies for the patient selection domain, in three (60%) studies for the reference standard domain and in two (40%) studies for the index test domain and the flow and timing domain. Risk of bias was unclear in three (60%) of the studies for the index test domain and the flow and timing domain. Only one study had low or unclear risk of bias in all domains. There were high applicability concerns in three (60%) studies for both the patient selection and reference standard domains. There were low applicability concerns for the index test domain for all studies. Only one study had low applicability concerns for all domains.

The high risks of bias were primarily due to studies including only confirmed cases of PTB or having inappropriate exclusions, interpreting index tests with knowledge of the reference standard, patients in the same study receiving different reference standards and the use of low-quality reference standards such as sputum smear microscopy. The high applicability concerns were primarily due to studies which only included confirmed TB cases or those in which not all participants received a high-quality reference standard.

Table 4 shows the quality assessment for the study in children. The study had low risk of bias in the patient selection, index test and reproducibility domains. It had unclear risk of bias in the reference standard domain as it was unclear whether the reference standard results were interpreted without knowledge of the index test results and unclear risk of bias in the flow and timing domain as the interval between the index test and reference standard was unclear. The study had low applicability concerns in all three domains.

**Table 3. QUADAS-2 assessments for studies in adults.**

| Study | Risk of bias | | | | Applicability concerns | | |
|---|---|---|---|---|---|---|---|
| ADULTS | Patient selection | Index test | Reference standard | Flow and timing | Patient selection | Index test | Reference standard |
| Agostinis 2017 [34] | High | High | High | Unclear | High | Low | High |
| Babasa 2019 [35] | Low | Unclear | Unclear | Unclear | Low | Low | Low |
| Fentress 2020 [33] | High | High | Low | Unclear | High | Low | Low |
| Montuori 2019 [9] | High | Unclear | High | High | Low | Low | High |
| Wagih 2020 [36] | High | Unclear | High | High | High | Low | High |

Low = low risk/concern; High = high risk/concern; Unclear = unclear risk/concern.

**Table 4. QUADAS-2 assessments for the study in children, with additional risk of bias domain for reproducibility.**

| Study | Risk of bias | | | | | Applicability concerns | | |
|---|---|---|---|---|---|---|---|---|
| CHILDREN | Patient selection | Index test | Reference standard | Flow and timing | Reprodu-cibility | Patient selection | Index test | Reference standard |
| Heuvelings 2019 [37] | Low | Low | Unclear | Unclear | Low | Low | Low | Low |

Low = low risk/concern; High = high risk/concern; Unclear = unclear risk/concern.

### Diagnostic accuracy results

**Adults.** In four of the five studies, diagnostic accuracy data was available for individual LUS findings or combinations of findings. Of the 20 findings or combinations of findings mentioned in the four studies, only five were mentioned in more than one study and are presented here: subpleural nodule, lung consolidation, pleural effusion, miliary pattern and cavitation. In Babasa 2019 [35], data were only available for a combination of LUS findings (subpleural nodules or pleural effusion or consolidation or C-lines). Fentress 2020 [33] presents data on subpleural consolidation but describes this finding as "morphologically identical" to the subpleural nodules reported by Agostinis 2017 [34]. For convenience, subpleural consolidation will be referred to in this study as subpleural nodules.

The sensitivity of the subpleural nodule sign ranged from 72.5% to 100.0% in the four studies. None of the four studies used a microbiological reference standard for all patients, all studies used different reference standards or composite reference standards and three of the four studies gave different reference standards to a subset of patients. Sensitivity of pleural effusion and cavitation ranged from 7.8% to 24.0% and from 4.0% to 30.0%, respectively, in the four studies. Sensitivity of lung consolidation and miliary pattern varied from 46.7% to 80.4% and from 0.0% to 6.7%, respectively, in three studies. [Fig 2].

Montouri 2019 [9] was the only paper to report specificity data in adults. The specificity of subpleural nodule, lung consolidation, pleural effusion and cavitation were 66.7%, 25.3%, 74.5% and 89.3%, respectively. In Babasa 2019 [35], the sensitivity and specificity of the combined 4 signs were 55.9% and 93.1%, respectively [Table 5].

**Children.** Heuvelings 2019 [37] reported diagnostic accuracy data for the following LUS findings: interrupted pleural line, consolidation, pleural gap, >3 B lines per intercostal space in more than two lung areas, pleural effusion and enlarged mediastinal lymph nodes. Children with unconfirmed TB (clinical diagnosis for PTB but negative microbiological test results) were considered positive based on the *a priori* acceptance of any reference standard in children. The sensitivity and specificity of the six LUS findings in Heuvelings 2019 [37] is shown in Table 6.

### Reproducibility results

Reproducibility data was only available for the study in children. The kappa scores for inter-rater reliability for five LUS signs are shown in Table 7.

### Discussion

The advent of affordable, portable ultrasound devices has led to the increasing use of POCUS for the diagnosis of a range of infectious diseases [17]. For adult pneumonia, meta-analyses suggest that LUS has similar or higher sensitivity and specificity to CXR [20–22]. For adult PTB, CXR has a sensitivity of 87% and a specificity of 89% [7]. If POCUS could be shown to have similar diagnostic accuracy to CXR for PTB, it would be an important finding. CXR is

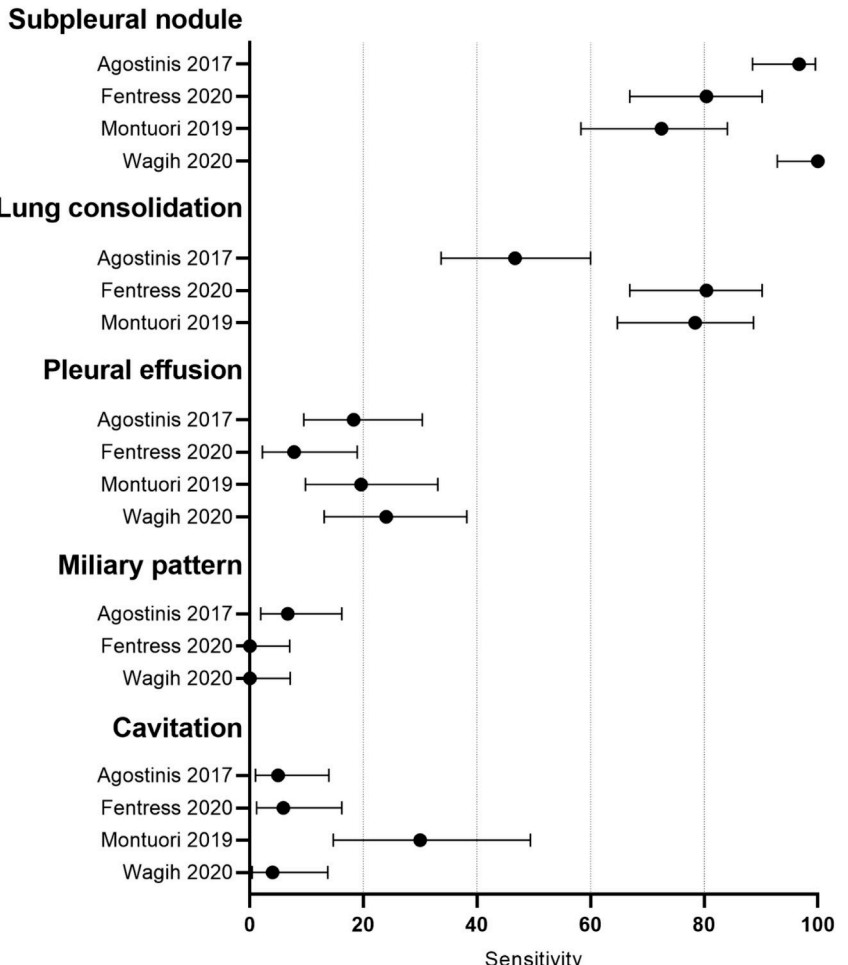

**Fig 2. Forest plot showing sensitivity of different lung ultrasound findings in adults.**

recommended by the WHO as a systematic screening tool for active PTB and as a triage tool to identify the patients with presumptive PTB who should be referred for confirmatory testing with culture or a molecular test [5,6]. However, access to CXR is limited in many low-income settings due to high hardware costs and paucity of skilled radiological staff [9,10,38]. POCUS devices are cheaper, safer and more portable than CXR. They can be powered by rechargeable batteries and inexpensive ultrasound gels produced using cornstarch and water or cassava flour, salt and water have been shown to produce comparable image quality to commercial ultrasound gels [39,40]. Additionally, POCUS can be performed by non-radiologists. For EPTB, FASH can be quickly taught to physicians with no prior ultrasound experience [41]. A similar tool for PTB could be valuable in resource-limited settings.

The results from this systematic review show that there is insufficient evidence to judge the diagnostic accuracy of POCUS for PTB. Only five adult studies were included in our review and all suffer from methodological limitations, including around patient selection, selection of an appropriate reference standard and blinding of reference standard results before application of ultrasound.

POCUS for PTB requires three stages. First, a defined image acquisition protocol should specify where the ultrasound probe should be pointed. Second, a defined image analysis

**Table 5. Diagnostic accuracy of lung ultrasound findings in adults.**

| Study | LUS finding | Patients | PTB | Sensitivity (95% CI) | Specificity (95% CI) |
|---|---|---|---|---|---|
| Agostinis 2017 [34] | Subpleural nodule | 60 | 60 | 96.7 (88.5–99.6) | |
| | Lung consolidation | 60 | 60 | 46.7 (33.7–60.0) | |
| | Pleural effusion | 60 | 60 | 18.3 (9.5–30.4) | |
| | Miliary pattern | 60 | 60 | 6.7 (1.9–16.2) | |
| | Cavitation | 60 | 60 | 5.0 (1.0–13.9) | |
| Babasa 2019* [35] | Subpleural nodules or pleural effusion or consolidation or C-lines | 131 | ** | 55.8 (45.7–65.5) | 92.3 (75.7–99.1) |
| Fentress 2020 [33] | Subpleural nodule | 51 | 51 | 80.4 (66.9–90.2) | |
| | Lung consolidation | 51 | 51 | 80.4 (66.9–90.2) | |
| | Pleural effusion | 51 | 51 | 7.8 (2.2–18.9) | |
| | Miliary pattern | 51 | 51 | 0.0 (0.0–7.0) | |
| | Cavitation | 51 | 51 | 5.9 (1.2–16.2) | |
| Montuori 2019 [9] | Subpleural nodule | 102 | 51 | 72.5 (58.3–84.1) | 66.7 (52.1–79.2) |
| | Lung consolidation | 102 | 51 | 78.4 (64.7–88.7) | 35.3 (22.4–49.9) |
| | Pleural effusion | 102 | 51 | 19.6 (9.8–33.1) | 74.5 (60.4–85.7) |
| | Miliary pattern | | | | |
| | Cavitation | 58 | 30 | 30.0 (14.7–49.4) | 89.3 (71.7–97.7) |
| Wagih 2020 [36] | Subpleural nodule | 50 | 50 | 100.0 (92.9–100.0) | |
| | Lung consolidation | | | | |
| | Pleural effusion | 50 | 50 | 24.0 (13.1–38.2) | |
| | Miliary pattern | 50 | 50 | 0.0 (0.0–7.1) | |
| | Cavitation | 50 | 50 | 4.0 (0.4–13.7) | |

95% CI = exact binomial 95% confidence interval; LUS = lung ultrasound.

*raw numbers of true positives, false positive, false negatives and true negatives were unavailable; confidence intervals are as presented in abstract so may not correspond to other calculated confidence intervals.

**number of patients with PTB not reported.

protocol should specify which LUS findings (e.g. subpleural nodule) should be detected. Finally, a defined image interpretation protocol should specify which combination of LUS findings provides the optimal balance of sensitivity and specificity to correlate with PTB.

In adults, two of the five included studies reported using the same image acquisition protocol, one study reported using a different image acquisition protocol and two studies did not provide this information. The lack of a consistent image acquisition protocol for POCUS for PTB contrasts with the use of POCUS for HIV-associated EPTB, for which the FASH image acquisition protocol was established in 2012 [19].

**Table 6. Diagnostic accuracy of lung ultrasound findings in children.**

| Study | LUS finding | n | PTB | Sensitivity (95% CI) | Specificity (95% CI) |
|---|---|---|---|---|---|
| Heuvelings 2019 [37] | Interrupted pleural line | 170 | 125 | 78.4 (70.2–85.3) | 26.7 (14.6–41.9) |
| | Consolidation | 170 | 125 | 45.6 (36.7–54.8) | 53.3 (37.9–68.3) |
| | Pleural gap | 170 | 125 | 52.8 (43.7–61.8) | 57.8 (42.2–72.3) |
| | >3 B lines | 170 | 125 | 28.0 (20.3–36.7) | 77.8 (62.9–88.8) |
| | Pleural effusion | 170 | 125 | 16.8 (10.7–24.5) | 91.1 (78.8–97.5) |
| | Enlarged lymph nodes | 116 | 84 | 19.0 (11.3–29.1) | 71.9 (53.3–86.3) |

95% CI = exact binomial 95% confidence interval; LUS = lung ultrasound.

**Table 7. Kappa scores for inter-rater reliability of lung ultrasound findings in children.**

| Study | LUS finding | Kappa score |
|---|---|---|
| Heuvelings 2019 [37] | Interrupted pleural line | 0.62 |
| | Consolidation | 0.84 |
| | >3 B lines | 0.73 |
| | Pleural effusion | 0.89 |
| | Enlarged lymph nodes | 0.56 |

LUS = lung ultrasound.

Similarly, no image analysis protocol for POCUS for PTB has emerged, as 76% (16/21) of the LUS findings or combinations of findings reported in the included studies were unique to one study. This also contrasts with POCUS for EPTB as the FASH protocol specifies the ultrasound findings (e.g. abdominal lymph nodes) which should be detected [19].

For image interpretation, evidence from this review suggests that the LUS findings of subpleural nodule and lung consolidation had the highest sensitivities, ranging from 72.5% to 100.0% and 46.7% to 80.4%, respectively. However, only Montuori 2019 [9] reported specificity data for individual LUS findings, finding specificities of 66.7% and 35.3% for subpleural nodule and lung consolidation, respectively. Babasa 2019 [35] reported sensitivity of 55.8% and specificity of 92.3% for the composite finding of subpleural nodules *or* pleural effusion *or* consolidation *or* C-lines. Montuori 2019 [9] reported that a composite finding of subpleural nodule *and* the precise finding of apical consolidation would give sensitivity of 31% and specificity of 96%, while a composite finding of subpleural nodule *or* apical consolidation would give sensitivity of 86% and specificity of 63%. However, 14% (7/51) of PTB patients in Montuori 2019 [9] were diagnosed on a clinical or radiological basis alone, with no bacteriological confirmation. Fentress 2020 [33] reported that a composite finding of subpleural nodule or lung consolidation had a sensitivity of 96% but did not report specificity data. The sensitivity of cavitation was low in all studies and is a clear limitation of POCUS. In Fentress 2020 [33], cavitation was detected in 51% of patients by CXR but only 6% by POCUS. However, Fentress 2020 [33] reported that no patients with cavitary disease would have been missed by a composite finding of subpleural nodule or lung consolidation. No data were available on the inter-observer or intra-observer reproducibility of POCUS findings in adults. Overall, there is not enough evidence to recommend an image interpretation protocol of POCUS findings for PTB.

High-quality studies minimising methodological flaws are required to further assess the use of POCUS for PTB in adults. Recommendations for the design of such studies are shown in Table 8.

Child PTB differs radiologically from adult PTB, with lymphadenopathy found in 83–96% of children with PTB and 10–43% of adults with PTB [42], and the penetration of ultrasound can examine the entire chest, unlike in adults [43], so the optimum image acquisition protocol is likely to differ for children. A protocol for imaging mediastinal lymphadenopathy in child PTB was proposed in 2017 [44] and was used in combination with a protocol designed for paediatric pneumonia [45] in the only paediatric study identified for inclusion in this review. A composite of several LUS findings may provide acceptable diagnostic accuracy, given the general difficulty in diagnosing child PTB. However, more studies in children are required.

Strengths of this systematic review included a comprehensive literature search, detailed data on the characteristics of each study, including the reference standards used, the setting and the level of training of the sonographer, providing an overall picture of how POCUS has been used for PTB in practice. Limitations are that we restricted our search to articles in

**Table 8. Recommendations for the design of a diagnostic accuracy study of POCUS for PTB in adults.**

| Domain | Characteristics |
|---|---|
| Study design | One-group prospective cross-sectional study of consecutive patients presenting with symptoms suggestive of PTB |
| | All patients given both POCUS exam and a uniform appropriate reference standard (see below) at approximately the same time (maximum within one week) |
| | Clinician giving POCUS exam blinded to the reference standard results and vice versa |
| Reference standard | Liquid or solid culture or a WHO-approved molecular test |
| POCUS protocols | Image acquisition protocol clearly defined |
| | Image analysis protocol for lung ultrasound findings (e.g. subpleural nodule) clearly defined |
| Diagnostic accuracy | Sensitivity and specificity calculated for each lung ultrasound finding |
| Reproducibility | Reproducibility of each lung ultrasound finding calculated between humans or between artificial intelligence and humans and a Kappa score calculated |
| Data sharing | Anonymised individual patient data made freely available so the optimal combination of lung ultrasound findings to correlate with PTB can be devised and compared between studies |
| | Images and metadata collected in a sharable manner |

English and French and did not include grey literature beyond those indexed in the databases we searched. Publication bias is another concern that we cannot rule out.

POCUS is an inexpensive, portable technology which could be a valuable diagnostic tool for PTB in resource-limited settings. This systematic review demonstrates that there is insufficient evidence to judge the diagnostic accuracy of POCUS for PTB. The current evidence base is limited and suffers from methodological flaws. Variability in the sensitivity of LUS findings between studies may be due to variable reference standards or may imply operator dependence. There is no consensus on the optimum image acquisition or image analysis protocols for POCUS for PTB. It is also not yet clear where POCUS fits in the diagnostic pathway for PTB.

The WHO's target product profile (TPP) for a triage tool for PTB suggests a minimum requirement of 90% sensitivity and 70% specificity with a price per test of less than $2 and a time to result of less than 30 minutes [28]. POCUS meets the cost and speed requirements of this TPP. New diagnostic accuracy studies which minimise potential sources of bias may show POCUS to be a viable triage test for PTB.

## Supporting information

**S1 Appendix. Search strategies.**
(PDF)

**S2 Appendix. Data extraction form.**
(PDF)

**S3 Appendix. Additional reproducibility domain for quality assessment.**
(PDF)

**S4 Appendix. QUADAS-2 guidance.**
(PDF)

**S5 Appendix. Reasons for exclusion at full-text screening.**
(PDF)

**S1 Table. PRISMA 2009 checklist.**
(DOC)

## Author Contributions

**Conceptualization:** Jacob Bigio, Madhukar Pai.

**Data curation:** Jacob Bigio, Mikashmi Kohli.

**Investigation:** Jacob Bigio, Mikashmi Kohli, Joel Shyam Klinton, Emily MacLean.

**Methodology:** Jacob Bigio, Mikashmi Kohli, Genevieve Gore.

**Project administration:** Jacob Bigio.

**Supervision:** Madhukar Pai.

**Visualization:** Jacob Bigio.

**Writing – original draft:** Jacob Bigio.

**Writing – review & editing:** Jacob Bigio, Mikashmi Kohli, Joel Shyam Klinton, Emily MacLean, Genevieve Gore, Peter M. Small, Morten Ruhwald, Stefan Fabian Weber, Saurabh Jha, Madhukar Pai.

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
