## [Decision Letter · Decision Letter 0]

25 Mar 2021

PONE-D-21-07253

Diagnostic accuracy of point-of-care ultrasound for pulmonary tuberculosis: a systematic review

PLOS ONE

Dear Dr. Bigio,

Thank you for submitting your manuscript to PLOS ONE. After careful consideration, we feel that it has merit but does not fully meet PLOS ONE’s publication criteria as it currently stands. Therefore, we invite you to submit a revised version of the manuscript that addresses the points raised during the review process.

Please submit your revised manuscript. If you will need significantly more time than this to complete your revisions, please reply to this message or contact the journal office at plosone@plos.org. Please include the following items when submitting your revised manuscript:

We look forward to receiving your revised manuscript.

Kind regards,

Frederick Quinn

Academic Editor

PLOS ONE

Journal Requirements:

3. We note that this manuscript is a systematic review or meta-analysis; our author guidelines therefore require that you use PRISMA guidance to help improve reporting quality of this type of study. Please upload copies of the completed PRISMA checklist as Supporting Information with a file name “PRISMA checklist”.

Reviewers' comments:

Reviewer's Responses to Questions

**Comments to the Author**

1. Is the manuscript technically sound, and do the data support the conclusions?

Reviewer #1: Yes

Reviewer #2: Yes

2. Has the statistical analysis been performed appropriately and rigorously? 

Reviewer #1: Yes

Reviewer #2: Yes

3. Have the authors made all data underlying the findings in their manuscript fully available?

Reviewer #1: Yes

Reviewer #2: Yes

4. Is the manuscript presented in an intelligible fashion and written in standard English?

Reviewer #1: Yes

Reviewer #2: Yes

5. Review Comments to the Author

Reviewer #1: Thank you to read this important and value article. Authors wrote an interesting paper on important issue: use of ultrasound in Tuberculosis diagnosis.

I think that the paper is very well wrote and need only minor revisions.

Below my suggestions:

1. Introduction: Well don. Clarify better why is important use ultrasound in poor setting and with high burden of TB.In fact, "resource-limited settings are of special interest, as radiological equipment and expertise are scarce, or even absent, due to their high costs or poor maintenance. A focused assessment of extra-pulmonary TB has been also proposed" (see and cite Focused ultrasound to diagnose HIV-associated tuberculosis (FASH) in the extremely resource-limited setting of South Sudan: a cross-sectional study. BMJ Open. 2019 Apr 2;9(4):e027179. )

2. Methods and results: clear

3. Discussion: add better the role of US in LMIC (low middle income countries)...There is an increasing interest in employing chest ultrasound in low- and middle-income countries Chest ultrasound has a relatively steep learning curve: It is ionization-free and is increasingly available at reasonable costs. Moreover, it can be portable and operated with rechargeable batteries. Ultrasound gel, the only routine supply item needed, can easily be produced locally [26], thus making it an attractive option in resource-limited settings. On the other hand, inter-observer variability and diagnostic errors represent important pitfalls, and should be investigated specifically for TB. While the conditions for a wider implementation are favorable, none of the studies were performed in low-income countries. This observation, added to the paucity of available evidence, indicates that the use of CUS for the diagnosis of thoracic TB is still a clinical niche. Also future perspectives of Ultrasound Tb in diagnosis of abdominal tb (see and cite Focused ultrasound to diagnose HIV-associated tuberculosis (FASH) in the extremely resource-limited setting of South Sudan: a cross-sectional study. BMJ Open. 2019 Apr 2;9(4):e027179d and see and cite Uncommon testicular localization of Disseminated TB: a case report from Mozambique. New Microbiol. 2019 Jul;42(3):184-187). For these reasons improve knowledge of POCUS is important to fight and tb burden control.

Furthermore, the role of age can be discuss in the TB presentation. Elderly can have different clinical presentation and outcome and for Ultrasound it is important know the difference between young and elderly people (see and citeActive Pulmonary Tuberculosis in Elderly Patients: A 2016-2019 Retrospective Analysis from an Italian Referral Hospital. Antibiotics (Basel). )

Reviewer #2: Thank you for a well-written paper on an emerging and important topic.

Please clarify some minor issues:

- Please indicate how many (if any) studies were not included due to them not being written in English or French. Also give references if applicable.

- S4 Appendix (QUADAS-2 guidance), Domain 2 (Index test), Signaling question 2 states that ultrasound scans do not have a threshold. I tend to disagree with this as positivity of an ultrasound scan might differ e.g. number/extent of B-lines, size of subpleural nodules/consolidation etc. Please clarify.

- Please include a table of excluded studies - with study reference and reason for exclusion. This can be supplementary, but will allow the reader to make a judgement on whether studies were appropriately excluded.

- Page 8, line 200 states that 'The high risks of bias were primarily due to case-control designs or inappropriate exclusions'. Please clarify as the study designs of all studies included (Table 1 & 2) were stated as prospective cross-sectional.

6. PLOS authors have the option to publish the peer review history of their article (what does this mean?). If published, this will include your full peer review and any attached files.

Reviewer #1: **Yes: **Francesco Di Gennaro

Reviewer #2: **Yes: **Daniël J. van Hoving

---

## [Author Response · Author response to Decision Letter 0]

13 Apr 2021

5. Review Comments to the Author

We thank the reviewers for their helpful comments.

Reviewer #1: Thank you to read this important and value article. Authors wrote an interesting paper on important issue: use of ultrasound in Tuberculosis diagnosis.

I think that the paper is very well wrote and need only minor revisions.

Below my suggestions:

1. Introduction: Well don. Clarify better why is important use ultrasound in poor setting and with high burden of TB.In fact, "resource-limited settings are of special interest, as radiological equipment and expertise are scarce, or even absent, due to their high costs or poor maintenance. A focused assessment of extra-pulmonary TB has been also proposed" (see and cite Focused ultrasound to diagnose HIV-associated tuberculosis (FASH) in the extremely resource-limited setting of South Sudan: a cross-sectional study. BMJ Open. 2019 Apr 2;9(4):e027179.

Lines 65-67 have been rephrased to emphasise that availability of CXR equipment and radiological staff is limited in many high TB-burden, low-resource settings. The sentence in line 73-75 has been extended to add that POCUS devices are attractive in low resource settings due to their low cost, portability and lack of requirement for radiological staff. Mention of the FASH protocol has been added in lines 78-80. We cited the Cochrane Review of abdominal ultrasound for EPTB, which includes the article from Bobbio et al in South Sudan suggested here. We have additionally added a citation to the Heller et al (2012) paper which originally proposed the FASH protocol.

2. Methods and results: clear

3. Discussion: add better the role of US in LMIC (low middle income countries)...There is an increasing interest in employing chest ultrasound in low- and middle-income countries Chest ultrasound has a relatively steep learning curve: It is ionization-free and is increasingly available at reasonable costs. Moreover, it can be portable and operated with rechargeable batteries. Ultrasound gel, the only routine supply item needed, can easily be produced locally [26], thus making it an attractive option in resource-limited settings. On the other hand, inter-observer variability and diagnostic errors represent important pitfalls, and should be investigated specifically for TB. While the conditions for a wider implementation are favorable, none of the studies were performed in low-income countries. This observation, added to the paucity of available evidence, indicates that the use of CUS for the diagnosis of thoracic TB is still a clinical niche. Also future perspectives of Ultrasound Tb in diagnosis of abdominal tb (see and cite Focused ultrasound to diagnose HIV-associated tuberculosis (FASH) in the extremely resource-limited setting of South Sudan: a cross-sectional study. BMJ Open. 2019 Apr 2;9(4):e027179d and see and cite Uncommon testicular localization of Disseminated TB: a case report from Mozambique. New Microbiol. 2019 Jul;42(3):184-187). For these reasons improve knowledge of POCUS is important to fight and tb burden control. Furthermore, the role of age can be discuss in the TB presentation. Elderly can have different clinical presentation and outcome and for Ultrasound it is important know the difference between young and elderly people (see and citeActive Pulmonary Tuberculosis in Elderly Patients: A 2016-2019 Retrospective Analysis from an Italian Referral Hospital. Antibiotics (Basel). )

The first paragraph of the discussion has been expanded to emphasise the potential utility of POCUS in LMICs in place of CXR due to high CXR hardware costs and lack of skilled radiological staff and the Bobbio et al paper in South Sudan has been cited. Mention of the possibility of producing ultrasound gel from inexpensive materials has been added, along with two references to studies comparing image quality between commercial and homemade ultrasound gels. Mention of rechargeable batteries has also been added. Mention of FASH being quick to teach to physicians with no prior ultrasound experience has been added, along with a reference to the short FASH curriculum paper from Heller et al 2010. FASH is additionally mentioned in lines 323-5 and 328-31, to contrast the established protocols for POCUS for EPTB with the lack of such protocols for POCUS for PTB. A note has been added in lines 347-8 to say that no data were available on the inter-observer or intra-observer reproducibility of POCUS findings in adults and recommendations for future reproducibility studies are given in table 8.

Reviewer #2: Thank you for a well-written paper on an emerging and important topic.

Please clarify some minor issues:

- Please indicate how many (if any) studies were not included due to them not being written in English or French. Also give references if applicable.

No studies were excluded for not being in English or French and this has now been indicated in the results.

- S4 Appendix (QUADAS-2 guidance), Domain 2 (Index test), Signaling question 2 states that ultrasound scans do not have a threshold. I tend to disagree with this as positivity of an ultrasound scan might differ e.g. number/extent of B-lines, size of subpleural nodules/consolidation etc. Please clarify.

We agree with you. We’ve changed the guidance in S4 Appendix based in part on the guidance in the Cochrane Review of abdominal ultrasound for TB. It now reads: 

“We will answer ‘yes’ if the study states the use of a single, pre-specified, cut-off value for each lung ultrasound sign for which it is appropriate (e.g. “subpleural, nodular, hypoechoic region < 1 × 1 cm, with distinct borders and trailing comet-tail artifacts”) and the study pre-specifies how each non-numerical sign was defined (e.g. "diffuse, bilateral pattern of multiple B-lines and subpleural sonographic granularity"). We will answer ‘no’ if multiple cut-off values or sign definitions were evaluated for any one sign and an optimal one was subsequently chosen based on maximising test accuracy. We will judge ‘unclear’ if any one cut-off value or sign definition was used but not reported or if we cannot tell.”

This is a more stringent criteria than in the Cochrane Review (as every cut-off value/sign definition has to be prespecified to be low risk, and any one unclear will render it unclear risk). However, we are happy to be stringent and for clarity prefer to present our QUADAS assessment for the index test domain as a single judgement of the whole study, rather than splitting it up by ultrasound finding as in the Cochrane Review.

With this change, risk of bias remains high in the index test domain for Agostinis and Fentress (based on signalling question 1) and remains unclear for Wagih (unclear for signalling question 1 and for every LUS sign for signalling question 2). Risk of bias changes from low to unclear in the same domain for both Babasa (unclear for the single composite LUS sign in the study for signalling question 2) and Montuori (unclear in three of the four LUS signs in the study for signalling question 2, and low risk for the fourth). In children, risk of bias remains low for the index test domain in the only study. Table 3 has been updated accordingly, as have the descriptions of QUADAS findings in the results section.

- Please include a table of excluded studies - with study reference and reason for exclusion. This can be supplementary, but will allow the reader to make a judgement on whether studies were appropriately excluded.

A table of excluded studies with references and reasons for exclusion has been added in S6 Appendix.

- Page 8, line 200 states that 'The high risks of bias were primarily due to case-control designs or inappropriate exclusions'. Please clarify as the study designs of all studies included (Table 1 & 2) were stated as prospective cross-sectional.

This was incorrectly stated. High risk of bias was given in Domain 1 (Patient selection) Signaling question 2 if a case-control study design was used or if the study only included confirmed cases of PTB (as both can produce biases in sensitivity estimates). 3 of the 5 adult studies included only confirmed cases but none were case-control studies. The sentence has therefore been corrected to “The high risks of bias were primarily due to studies including only confirmed cases of PTB or having inappropriate exclusions…”

Additionally, in two places reference has been added to the WHO consolidated guidelines on tuberculosis. Module 2: screening – systematic screening for tuberculosis disease, which was released this week.

---

## [Decision Letter · Decision Letter 1]

23 Apr 2021

Diagnostic accuracy of point-of-care ultrasound for pulmonary tuberculosis: a systematic review

PONE-D-21-07253R1

Dear Dr. Bigio,

We’re pleased to inform you that your manuscript has been judged scientifically suitable for publication and will be formally accepted for publication once it meets all outstanding technical requirements.

Kind regards,

Frederick Quinn

Academic Editor

PLOS ONE

Additional Editor Comments (optional):

Reviewers' comments:

Reviewer's Responses to Questions

**Comments to the Author**

1. If the authors have adequately addressed your comments raised in a previous round of review and you feel that this manuscript is now acceptable for publication, you may indicate that here to bypass the “Comments to the Author” section, enter your conflict of interest statement in the “Confidential to Editor” section, and submit your "Accept" recommendation.

Reviewer #1: All comments have been addressed

Reviewer #2: All comments have been addressed

2. Is the manuscript technically sound, and do the data support the conclusions?

Reviewer #1: Yes

Reviewer #2: Yes

3. Has the statistical analysis been performed appropriately and rigorously? 

Reviewer #1: Yes

Reviewer #2: Yes

4. Have the authors made all data underlying the findings in their manuscript fully available?

Reviewer #1: Yes

Reviewer #2: Yes

5. Is the manuscript presented in an intelligible fashion and written in standard English?

Reviewer #1: Yes

Reviewer #2: Yes

6. Review Comments to the Author

Reviewer #1: Authors Wrote an very interesting paper. Use of ultrasound in tb diagnosed is crucial for tb burden control

Reviewer #2: (No Response)

7. PLOS authors have the option to publish the peer review history of their article (what does this mean?). If published, this will include your full peer review and any attached files.

Reviewer #1: No

Reviewer #2: **Yes: **Daniël van Hoving

---

## [Editor Report · Acceptance letter]

29 Apr 2021

PONE-D-21-07253R1 

Diagnostic accuracy of point-of-care ultrasound for pulmonary tuberculosis: a systematic review 

Dear Dr. Bigio:

I'm pleased to inform you that your manuscript has been deemed suitable for publication in PLOS ONE. Congratulations! Your manuscript is now with our production department. 

Kind regards, 

on behalf of

Dr. Frederick Quinn 

Academic Editor

PLOS ONE